# Prevalence of depression among students at Ethiopian universities and associated factors: A systematic review and meta-analysis

**Tamrat Anbesaw**[1]*, **Yosef Zenebe**[1], **Mogessie Necho**[1], **Moges Gebresellassie**[2], **Tesfaye Segon**[3], **Fasikaw Kebede**[4], **Tilahun Bete**[5]

1 Department of Psychiatry, College of Medicine and Health Science, Wollo University, Dessie, Ethiopia, 2 Department of Psychology, Teachers Education and Behavioural Science, Wollo University, Dessie, Ethiopia, 3 Department of Psychiatry, College of Health Science, Mettu University, Metu, Ethiopia, 4 Department of Epidemiology, College of Health Science, Woldia University, Woldia, Ethiopia, 5 Department of Psychiatry, School of Nursing and Midwifery, College of Health and Medical Sciences, Haramaya University, Harar, Ethiopia

* tamratanbesaw@gmail.com

**Data Availability Statement:** All relevant data from this study will be made available upon study completion.

## Abstract

### Background

Depression is the most common cause of disability in the world, which affects 350 million people. University students struggle to cope with stressors that are typical of higher education institutions as well as anxiety related to education. Although evidence indicates that they have a high prevalence of depression, no reviews have been done to determine the prevalence of depression among students at Ethiopian universities comprehensively.

### Methods

Without regard to time constraints, PubMed, Scopus, and EMBASE were investigated. A manual search for an article reference list was also conducted. The Meta XL software was used to extract relevant data, and the Stata-11 meta-prop package was used to analyze it. The Higgs I2 test was used to test for heterogeneity.

### Results

A search of the electronic and manual systems resulted in 940 articles. Data were extracted from ten studies included in this review involving a total number of 5207 university students. The pooled prevalence of depression was 28.13% (95% CI: 22.67, 33.59). In the sub-group analysis, the average prevalence was higher in studies having a lower sample size (28.42%) than studies with a higher sample; 27.70%, and studies that utilized other (PHQ-9, HADS); 30.67% higher than studies that used BDI-II; 26.07%. Being female (pooled AOR = 5.56) (95% CI: 1.51, 9.61), being a first-year (pooled AOR = 4.78) (95% CI: 2.21, 7.36), chewing khat (pooled AOR = 2.83) (95% CI: 2.32, 3.33), alcohol use (pooled AOR = 3.12 (95% CI:3.12, 4.01) and family history of mental illness (pooled AOR = 2.57 (95% CI:2.00, 3.15) were factors significantly associated with depression.

**Funding:** The authors received no specific funding for this work.

**Competing interests:** The authors have declared that no competing interests exist.

**Abbreviations:** AOR, Adjusted Odds Ratio; BDI-I, Beck's Depression Inventory-I; CES-D, Center for Epidemiologic Studies Depression scale; CI, Confidence Interval; CS, Cross-Sectional; ETB, Ethiopian Birr; HADS, Hospital Anxiety and Depression Scale; OR, Odds Ratio; PHQ-9, Patient Health Questionnaire-9; PRISMA-P, Preferred Reporting Items for Systematic Reviews and Meta-analysis.

## Conclusion

This systematic review and meta-analysis revealed that more than one-fourth of students at Ethiopian universities had depression. More efforts need to be done to provide better mental healthcare to university students in Ethiopia.

## Background

Depression is a common mental disorder which is characterized by sadness, loss of pleasure or interest, disturbance of sleep, psychomotor activity, difficulty to concentrate, decreased energy, guilty feeling, and recurring thought of death wish [1]. Depression has received increasing global attention because of its negative effects on interpersonal, social, and occupational functioning [2].

University students are a special group of people who are going through a key transition from adolescence to adulthood, which may be one of the most stressful times in their lives. Many students experience anxiety as they try to fit in, keep decent grades, prepare for the future, and be away from home [3]. As a reaction to this stress, some students become depressed. Also, depression contributes to lower academic performance, the chance of dropping out, suicidal behavior (ideation, plan, and attempt), and impact on peer and teacher interactions in addition to negative health consequences [4]. Without recognizing depression, students may cry all the time, skip classes, or isolate themselves [5]. Globally, the prevalence of depression among university students is estimated differently in different studies from 1.4% to 73.5% [6]. Depression has a great impact or impairment among university students that require attention for a better existence in public society. Various Studies across the world have reported different prevalence rates for depression. The prevalence of depression among university students in the United States 27.2% [3], Pakistan 42.66% [7], Iran 33% [5], and China 74% [8].

Different factors significantly associated with depression among university students such as being single [5], female gender [9–13], age [11, 14, 15], low academic achievement [11, 16], family problems [11, 17, 18], poor social support [19], family history of mental illness [17, 20], parental education [11, 17, 21], financial struggles [14, 16], the field of study [22], year of study [13, 14, 19], type of college [11], the satisfaction of major study [17], risky sexual behavior [23, 24], and substance use (alcohol, tobacco, and khat) [11, 16, 20, 23, 25–27]. Studies conducted in Ethiopia showed, being female [25, 26, 28], being a first-year student [25, 26, 28–30], monthly pocket money [31], having a mentally ill family member [30, 31], stressful life events [29], violent behavior [31, 32], being from the College of Social science and humanity [31], younger age [29], having a chronic medical illness [30], and current use of illicit substances [27].

Even though a wide range of studies showed depression as a significant public health problem in developing nations including Ethiopia, still there is no systematic review and meta-analysis conducted to assess the prevalence of depression among university students. Therefore, this systematic review and meta-analysis aimed to summarize the existing evidence on the prevalence of depression and the pooled odds ratio of the associated factors for depressive symptoms among university students and to formulate possible suggestions for future clinical practice and research community.

## Materials and methods

### Study designed

The PRISMA (preferred reporting items for systematic reviews and meta-analyses) standard was used to perform the frame of the whole review process [33].

### Search strategy

An electronic and manual search of eligible articles was performed as part of a systematic review of the literature. Our search was conducted on October 10, 2022, using electronic libraries in Scopus, PubMed, and EMBASE, as well as manual exploration of the reference lists of articles. For searching articles on the prevalence of depression among university students using the PubMed database, we used the following search terms: "epidemiology" OR "prevalence" OR "magnitude" OR "incidence" AND "factor" OR "associated factor" OR "risk" OR "risk factor" OR "determinant", "depressive symptoms", "depressive disorder" OR "major depressive disorder" AND "University students AND Ethiopia". Besides, the literature search in EMBASE and Scopus followed database-specific searching parameters. Furthermore, there was no specification for studies based on the study period in the reference list of included studies.

### Inclusion and exclusion criteria

The researchers included original quantitative studies on the frequency and determinants of depression among university students. All observational studies were conducted by using different study-designed cross-sectional reports from June 2006 up to June 2021 were included. This systematic review and meta-analysis included publications with full-text papers and studies of depression that were published in peer-reviewed journals. Fortunately, studies published as review articles, qualitative studies, brief reports, letters to the editor, or editorial comments, working papers publications, published in a language other than English, research on non-human subjects, and studies with duplicate data from other studies were also excluded.

### Outcome measurements

We have two objectives in this systematic review and meta-analysis study. These are to determine the pooled prevalence of depression among university students in Ethiopia and to estimate the pooled effects of associated factors with depression among university students in Ethiopia. The pooled prevalence of depression was calculated using STATA version 14.0. The pooled effect estimate of associated factors with depression was calculated. The odds ratio was prepared from the searched research reports using two by two tables.

### Data extraction and appraisal of study quality

Two authors (TA, and YZ) checked study titles and abstracts for eligibility after deleting duplicates. The full texts were evaluated by the same reviewers if at least one of them thought an article was potentially eligible. Two authors (TA and YZ) extracted detailed information using a Microsoft Excel spreadsheet after the papers were scrutinized for their titles, abstracts, and entire texts. All studies approved by both reviewers were included and any differences were worked out through discussion to reach a consensus. Following the agreement, information about the principal investigator, years of publications, study period, study population, and sample size was retrieved from the identified articles. The identified articles were organized using EndNote X7.3.1. Each of the included studies' risk of bias was assessed by six (TA, MN, MG, TS, FK, and TB) investigators. The Newcastle Ottawa quality evaluation checklist was

used to assess the quality of the studies included in the final analysis [34]. Study participants and setting, research design, recruiting technique, response rate, sample representativeness, valid measuring convention, measurement reliability, and proper statistical analysis are all included in the quality evaluation checklist.

## Data synthesis and analysis

We used a random-effect model to assess the overall prevalence of depression and the related variables for depression among university students, as well as their 95% CI's [35]. Meta-XL version 5.3 [36] was employed to extract relevant data from included studies and the STATA11 Meta-prop package [37] was implemented to estimate the pooled prevalence of depression among university students and pooled odds ratio of the associated factors for depression. The Higgs $I^2$ statistic was also utilized to detect heterogeneity. Thus, percentages $I^2$ statistical values around 0% ($I^2$ 0), 25% ($I^2$ 25), 50% ($I^2$ 50), and 75% ($I^2$ 75) would mean absent, low, medium, and high heterogeneity, respectively [38]. Subgroup analysis and sensitivity analysis analyses were also used to investigate the source of heterogeneity among the studies included. To detect publication bias, researchers utilized the funnel plot test [39] and the eggers publication bias test.

## Search results

### Identification of studies

Our search with the pre-specified search strategies resulted in an overall of 935 articles. Besides, five articles were obtained from the reference list of included articles making the total number of retrieved articles to be 930 [25, 27, 29, 30, 32]. Of this, we removed 45 duplicated studies before further screening. In the next stage, we excluded 895 by title screening, being irrelevant to the main subject; and repetitive publications. Therefore the remaining 23 articles had been completely inspected for eligibility to be included in the current systematic review and meta-analysis study; nevertheless, only 10 articles were tailored in the final meta-analysis since the rest 13 articles were also excluded due to various reasons; 7 articles were poor methodological assessment, 4 articles were reviewed studies, and 2 articles were published other than the English language) (Fig 1).

### Characteristics of included studies

We included ten studies that assessed the prevalence and associated factors of depression among university students [25–32, 40, 41]. These studies included a total of 5207 university students. Five [27, 30, 31, 40, 41], two [25, 28] two [26, 32], and one [29] of the included studies used the BDI-II, PHQ-9, CESD's questionnaire, and HADS, respectively, to measure depression in university students. Regarding the study's design, all studies were institutional-based cross-sectional [25–32, 40, 41]. Also, eight of the studies employed a simple random sampling technique during data collection, and two studies used systematic random sampling [27, 29]. All of the studies reported response rates [25–32, 40, 41] (Table 1).

### Quality of included studies

The quality of ten studies [25–32, 40, 41] was assessed with the modified Newcastle Ottawa quality assessment scale. This scale divides the total quality score into 3 ranges; a score of 7 to 10 as very good/good, a score of 5 to 6 as having satisfactory quality, and a quality score less than 5 as unsatisfactory [42]. All studies had scored good quality (Table 2).

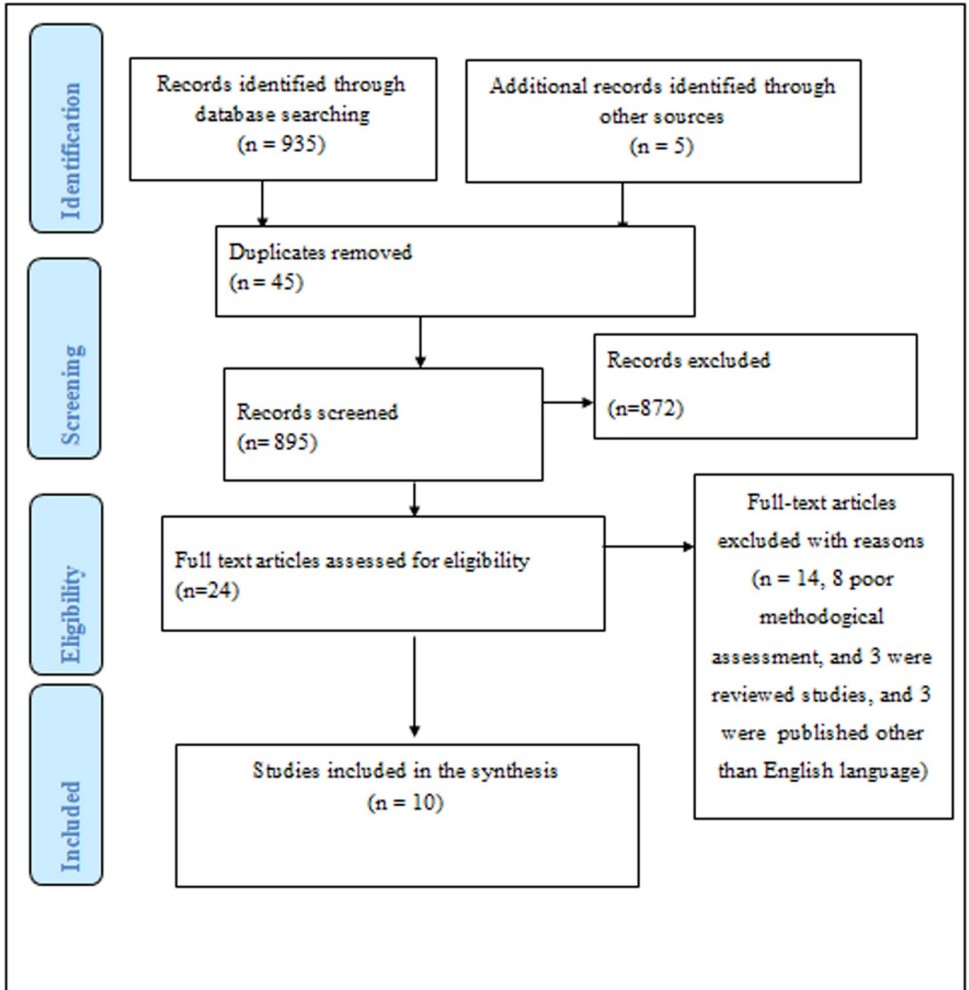

**Fig 1. PRISMA flow chart for the review search process.**

## The pooled prevalence of depression among university students

The pooled prevalence of depression among students at Ethiopian universities was found to be 28.13% (95% CI: 22.67, 33.59); ($I^2$ = 100%, p-value < 0.001) (Fig 2).

## Subgroup analysis of the prevalence of depression among university students

**Subgroup analysis of the prevalence of depression among university students by the sample size.** Since the average prevalence of depression was significantly influenced by the difference between the included studies, it was mandatory to conduct a subgroup analysis. Therefore, we used a sample size of below 400 [25, 27, 31, 32] provided a higher prevalence of depression; 28.42% than those who had a sample size above 400 [25, 26, 29, 30, 40, 41]; 27.70% (Fig 3).

**Subgroup analysis of the prevalence of depression among university students by the tools.** The measurement tools for depression to perform subgroup analysis. The subgroup analysis by assessment instrument yields that measurement with other (PHQ-9, HADS) [26,

**Table 1. Characteristics of studies on depression among university students are incorporated in this meta-analysis according to author's first name, year of publication, setting of the study, design, sample size, assessment tools, study population, sampling methods, age, and magnitude of depression, response rate.**

| Author, year of publication | Place | Study design | Sample size | Instrument and cutoff value | Study Population | Sampling Technique | Age | Overall prevalence of depression (%) | Response Rate |
|---|---|---|---|---|---|---|---|---|---|
| Birhanu et al. 2016 [25] | Ambo, Ethiopia | CS | 410 | CES-D >22 | University students | Simple random | M & F 17–23 years | 32.2.(n = 132) | 96.9%. |
| Tamene et al. 2021 [26] | Debre birhan, Ethiopia | CS | 369 | PHQ-9 ≥10 | University students | Simple random | M & F 18 and 33 | 17.1.(n = 63) | 100% |
| Ahmed et al. 2020 [31] | Jimma, Ethiopia | CS | 556 | BDI-II> 14 | University students | Simple random | M & F 18 to 35 years | 28.2.(n = 157) | 94.9% |
| Terasaki et al. 2009 [32] | Hawassa, Ethiopia | CS | 1,176 | PHQ-9 >10. | University students | Simple random. | M & F > 18 years | 23.6.(n = 277) | 100% |
| Kebede et al. 2019 [29] | Addis Abba, Ethiopia | CS | 273 | HADS>8 | University students | Systematic sampling | M & F 18–21 years | 51.30(140) | 98.5% |
| Muhammed et al. 2019 [30] | Wollo, Ethiopia | CS | *334* | BDI-II 14–63 | University students | Simple random | M & F 18 to 35 years | *35.3(118)* | 100% |
| Dagnew et al. 2020 [41] | Gondar, Ethiopia | CS | 383 | BDI-II = 21–63 | University students | Simple random | M & F 18–34 | 34.73(n = 133) | 97.7% |
| Worku et al. 2020 [40] | Arsi, Ethiopia | CS | 384 | BDI-II >14 | University students | Simple random | M & F 18–30 years | 4.4(n = 17) | 100% |
| Berhanu et al. 2020 [28] | Addis Ababa, Ethiopia | CS | 300 | CES-D >16 | University students | Simple random | M & F 17–28 years | 27.7(n = 83) | 95.5% |
| Teshome et al. 2020 [27] | Haramaya, Ethiopia | CS | 1022 | BDI-II > 13 | University students | Systematic random | M & F 20–24 | 26.8(274) | 98.3% |

**Key:** M = Male, F = Female

**Table 2. Quality assessment result of the studies included in this meta-analysis.**

| Study ID | Representation | Sampling | Random Selection | Non-response Bias | Data Collection | Case Definition | Reliability and Validity | Method of Data Collection | Prevalence Period | Numerator and Denominator | Summary |
|---|---|---|---|---|---|---|---|---|---|---|---|
| Birhanu et al. 2016 [25] | 1 | 1 | 1 | 1 | 1 | 1 | 1 | 1 | 1 | 1 | 10 |
| Tamene et al. 2021 [26] | 0 | 1 | 1 | 0 | 1 | 1 | 1 | 1 | 1 | 1 | 8 |
| Ahmed et al. 2020 [31] | 1 | 1 | 1 | 1 | 1 | 1 | 1 | 1 | 1 | 1 | 10 |
| Terasaki et al. 2009 [32] | 1 | 1 | 1 | 1 | 1 | 1 | 1 | 1 | 1 | 1 | 10 |
| Kebede et al. 2019 [29] | 0 | 1 | 1 | 1 | 0 | 0 | 1 | 1 | 1 | 1 | 7 |
| Muhammed et al. 2019 [30] | 0 | 1 | 1 | 1 | 1 | 0 | 1 | 1 | 0 | 1 | 7 |
| Dagnew et al. 2020 [41] | 0 | 1 | 1 | 1 | 1 | 1 | 1 | 1 | 1 | 1 | 9 |
| Worku et al. 2020 [40] | 0 | 1 | 1 | 1 | 1 | 1 | 1 | 1 | 1 | 1 | 9 |
| Berhanu et al. 2020 [28] | 0 | 1 | 1 | 1 | 1 | 1 | 1 | 1 | 1 | 1 | 9 |
| Teshome et al. 2020 [27] | 1 | 1 | 1 | 1 | 1 | 1 | 1 | 1 | 1 | 1 | 10 |

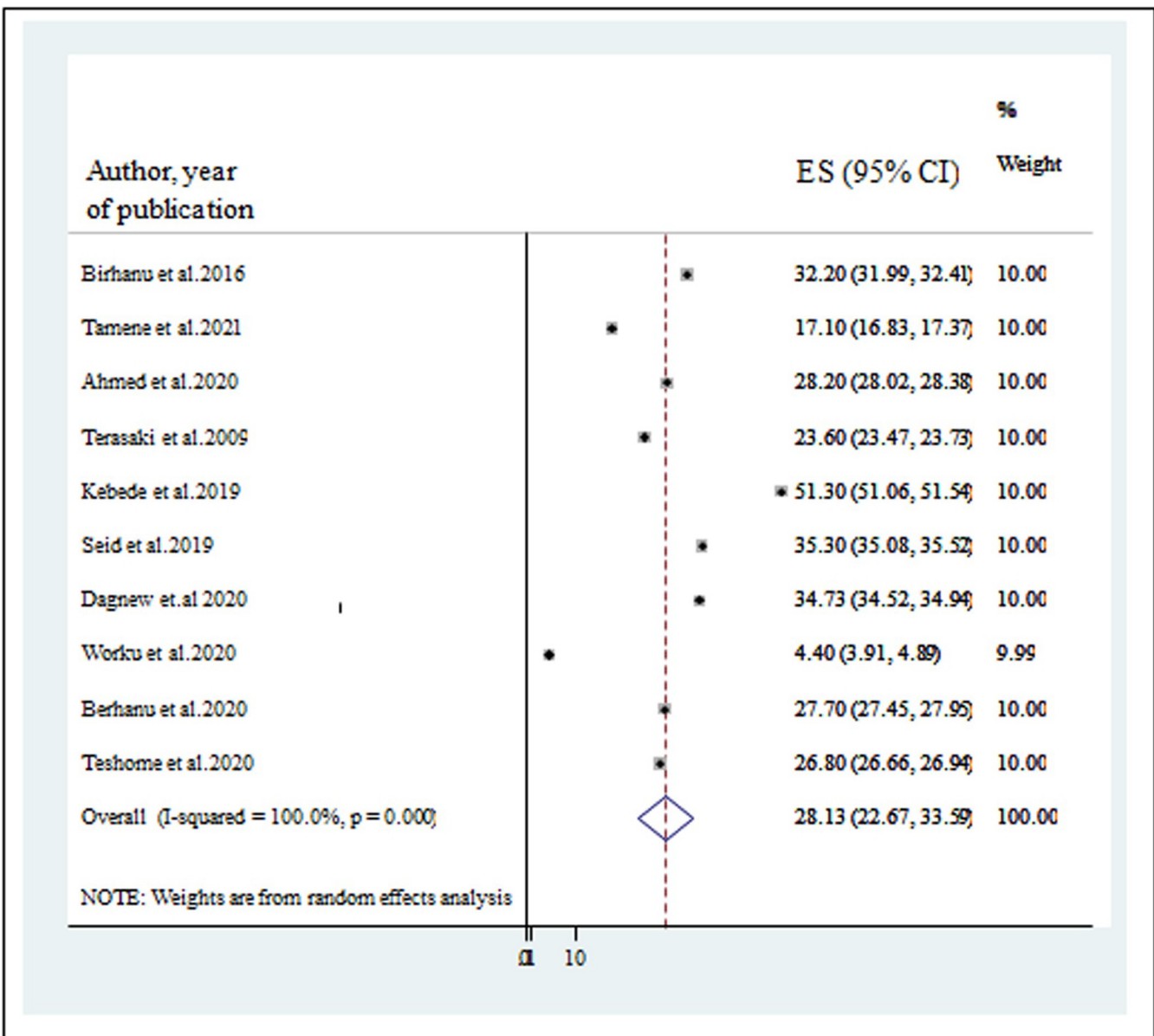

**Fig 2. A forest plot for the prevalence of depression.**

29, 32] provided relatively higher result, 30.67% (95% CI: 12.06, 49.27) with (I2 = 100%, p < 0.001) than the result with CEDS's [25, 27], which was 29.50% (95% CI:24.21, 34.79) (I2 = 99.9%, p < 0.001) and BDI-II [25, 30, 31, 40, 41], which was 26.07(19.42,32.72) (I2 = 100%) (Fig 4).

## Sensitivity analysis

The sensitivity analysis was performed to identify whether one or more of the ten studies had out-weighted the average prevalence of depression among university students. However, the findings show that all values are within the estimated 95% confidence interval, indicating that the absence of one study had no significant difference in the prevalence of this meta-analysis (Fig 5).

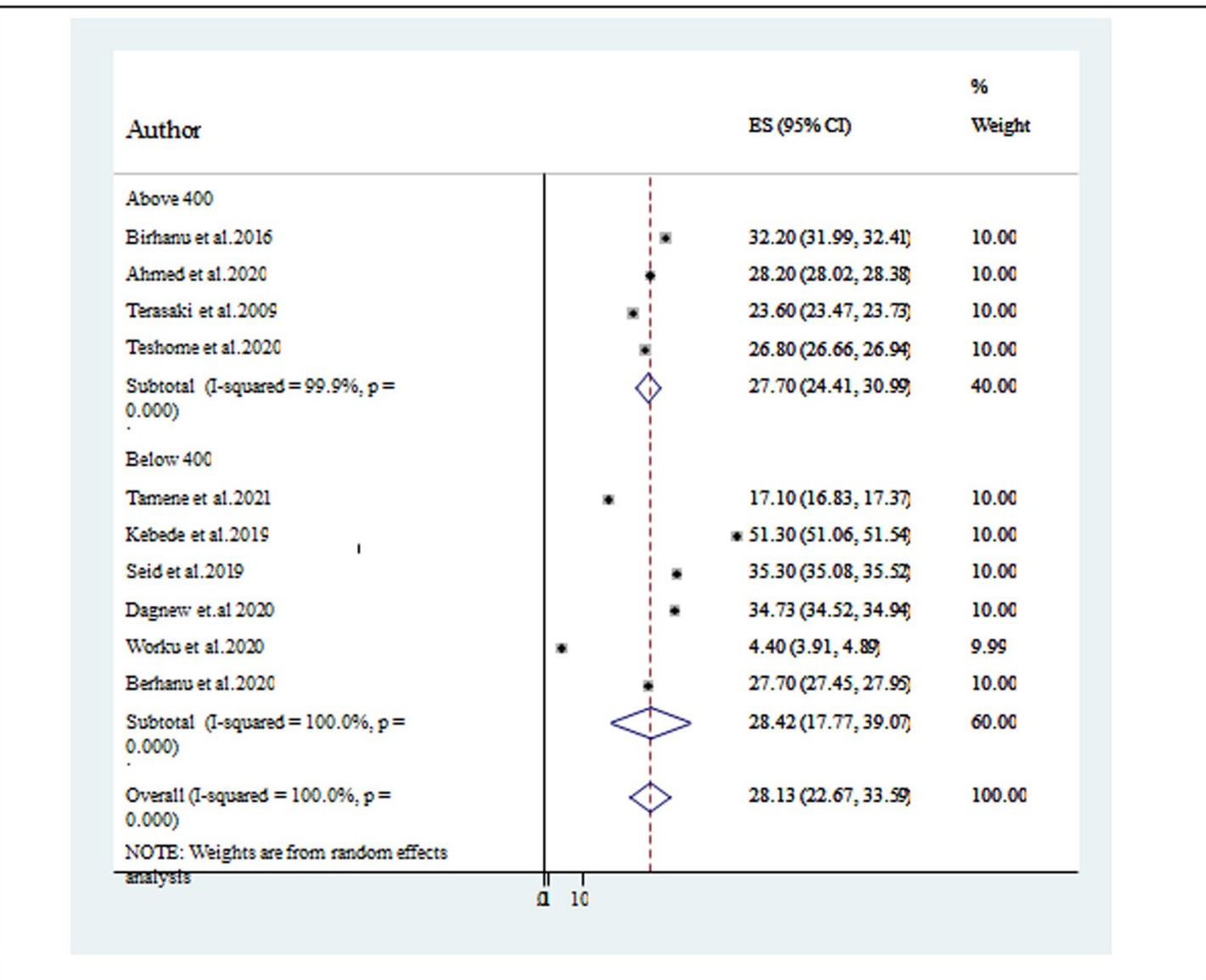

**Fig 3. A forest plot for the sub-group analysis of the prevalence of depression based on the sample size of studies.**

### Publication bias

A scatter plot of the logit event rate of depression on the X-axis and its standard error on the Y-axis was done, which showed that there was a publication bias since the graph was slightly asymmetrical. However, the Eggers publication bias test revealed that there was no significant publication bias (B = 9.19, SE = 94.5, and P-value = 0.92) (Fig 6).

### Associated factors for depression among students at Ethiopian universities

As stated previously, ten studies [25–32, 40, 41] reported one or more factors related to the development of depression among university students. Our narrative synthesis revealed that being female [25, 26, 28], being a first-year student [25, 27, 30], current use of khat [25, 30],

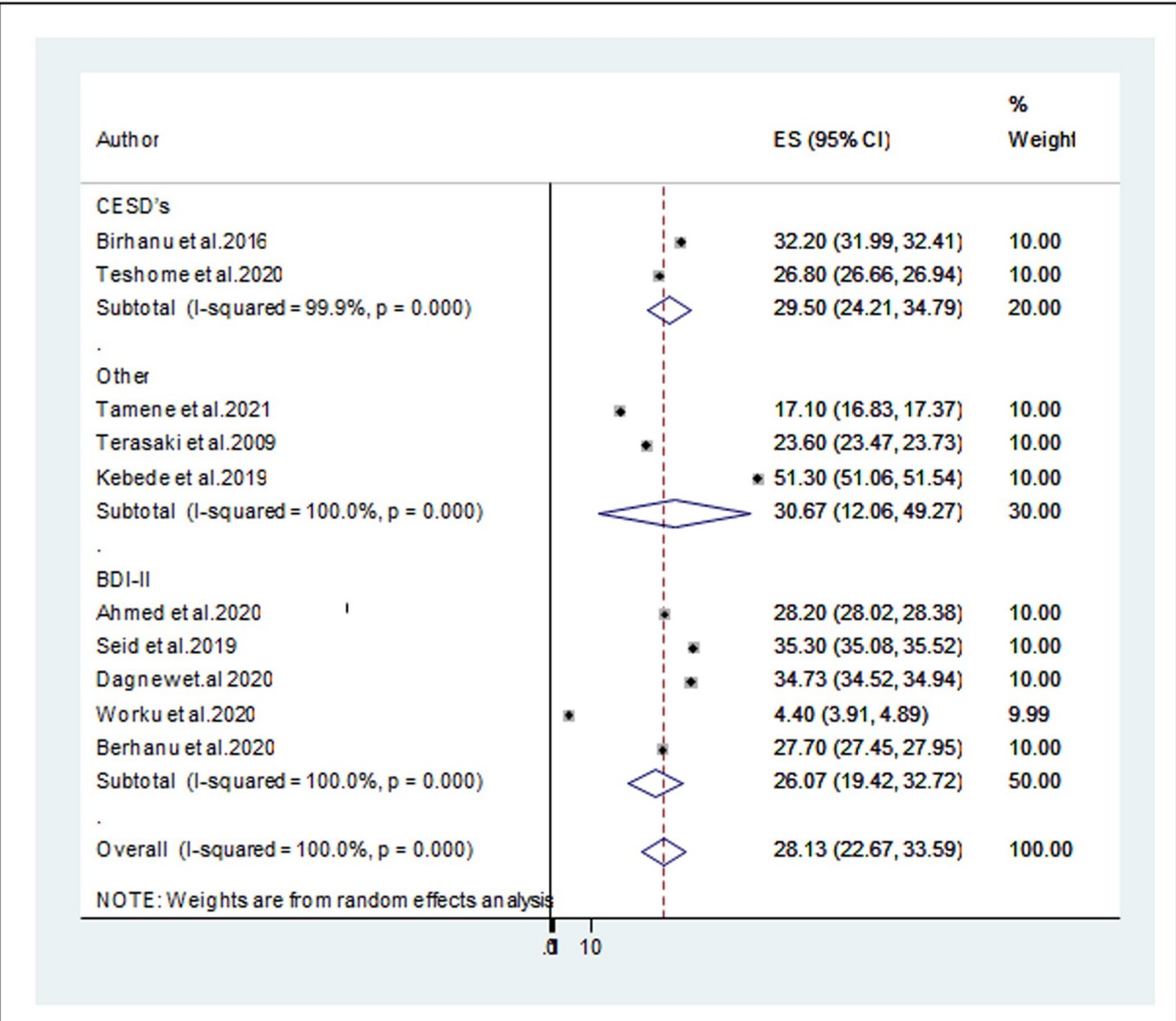

**Fig 4. Forest plot for the sub-group analysis of the prevalence of depression by measurement tool used.**

alcohol use [26, 27], and having a family history of mental illness [30, 31] were among the most commonly reported factors contributing to the development of depression among university students (Table 3).

The pooled odds ratio of being female among the above-mentioned studies was 5.56 (95% CI: 1.51, 9.61). This implied that female students were 5.56 times at higher risk of developing depression than male students. The pooled odds ratio for being a first-year student for the three studies reported above was found to be 4.78 (95% CI: 2.21, 7.36). Students who were first-year students were 4.78 times more likely to be depressed than senior students. History of

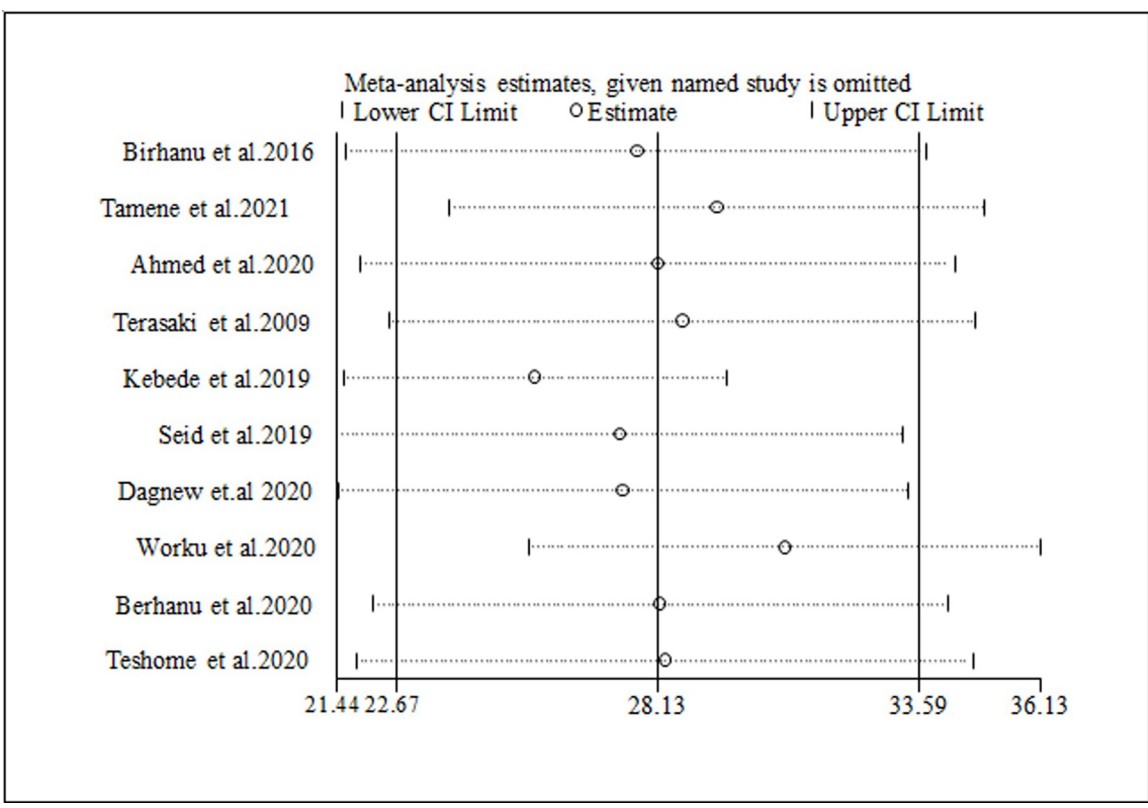

**Fig 5. Sensitivity analysis for the prevalence of depression among university students.**

chewing khat was also an associated factor for the development of depression with a pooled estimate odds ratio of 2.83 (95% CI: 2.32, 3.33). Besides, alcohol use was also found to have a significant association with the development of depression with an estimated pooled odds ratio of 3.12 (95% CI:3.12, 4.01). Participants who had a family history of mental illness was found to have a significant association with the development of depression with an estimated pooled odds ratio of 2.57 (95% CI:2.00, 3.15) (Table 4).

## Discussion

The pooled estimated prevalence of this systematic review and meta-analysis was 28.13% with a 95% CI (95% CI: 22.67, 33.59). This result was in line with another study conducted in China (32.74%) which analyzed 15 studies and 35,160 students [8]. It was also consistent with the result of a systematic review and meta-analysis study from Iranian university students which assessed 35 studies with a sample size of 9743 and 33% of them were found to have depression [5]. It was consistent with the study of Chinese university students which assessed 113 studies, and 28.4% of them were found to have depression [43]. Another study that involved 76,608 and 37 studies from low and middle-income countries [44] reported 24.4% of students as having depression, which was also supportive of the current finding. Our meta-analysis is much higher than in investigating the pooled prevalence of depression among the general population in Ethiopia (9.1% to 11%) [45, 46]. The findings revealed that several distinct characteristics of university students, such as increased social interactions and shifting residential and financial situations, may raise the risk of depression [47].

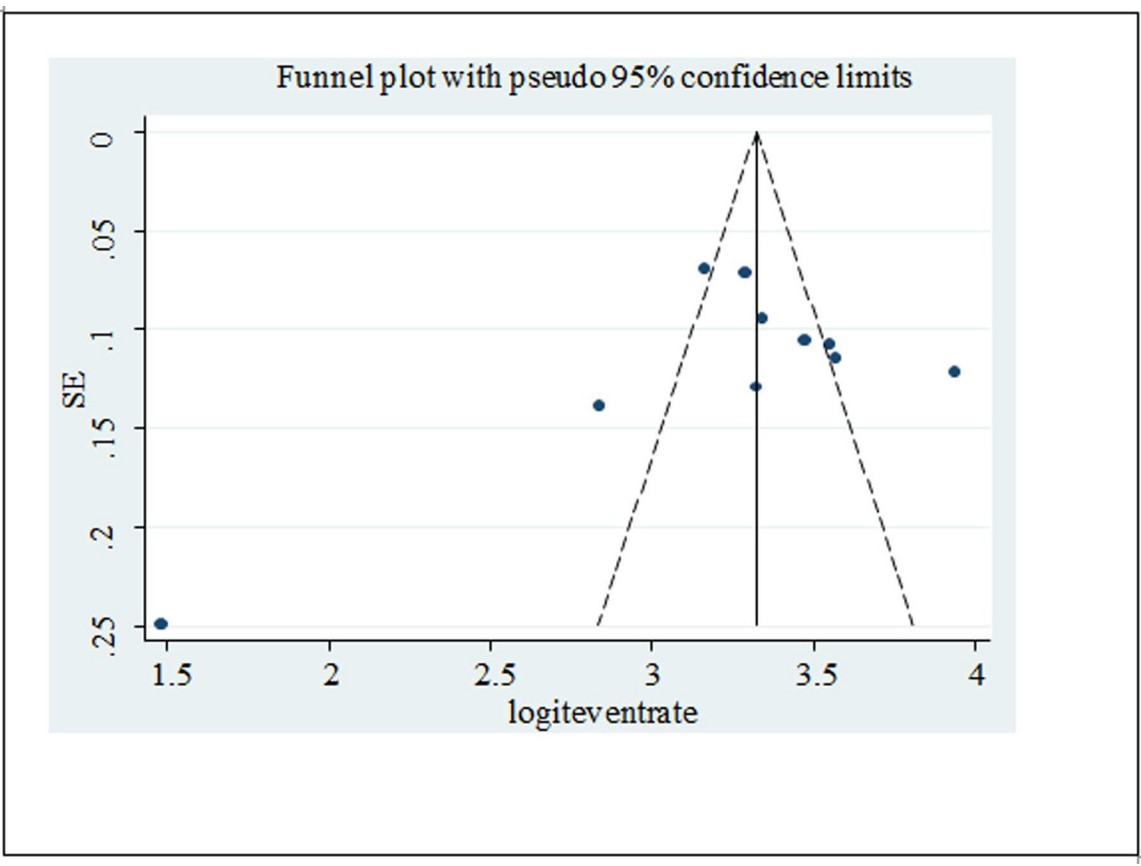

**Fig 6. A funnel plot for publication bias for depression.**

On the contrary, the average prevalence of depression in the present review was lower when compared with Asian university students on 8916 subjects, and in 27 articles a pooled prevalence of depression was 34.0% [48]. It was also lower than the systematic review and meta-analysis conducted on depression in Pakistani among 7652 university students and 26 studies in which the mean prevalence of depression was 42.66% (95% CI: 34.8–50.9%) [7]. The reason for the discrepancy might be because these investigations used different evaluation standards and measurement instruments, there could have been differences in prevalence rates.

The pooled prevalence of depression among university students in studies using a sample size below 400 study subjects (28.42%) [25, 27, 31, 32] was higher than the pooled prevalence of depression in university students that used a sample size of greater than 400 (27.07%) [25, 26, 29, 30, 40, 41]. The reason could be a smaller sample size increases the probability of a standard error thus providing a less precise and reliable result with weak power. Likewise, the present study revealed that pooled prevalence of depression was higher in studies as measured with other (PHQ-9, HADS) [26, 29, 32]; 30.67% (95% CI: 12.06, 49.27) than the result with CEDS's [25, 27] (29.50% (95% CI:24.21, 34.79) and BDI-II [25, 30, 31, 40, 41], which was 26.07 (19.42,32.72). This could be because studies that utilized delineated (PHQ-9, HADS) a lower cut-off point (PHQ-9 score ≥ 10 and HADS score > 8), which might result in an overestimation of the prevalence of depression.

Regarding the associated factors of depression, ten studies [25–32, 40, 41] had reported different factors and being female [25, 26, 28], being a first-year student [25, 27, 30], current use

**Table 3. Characteristics of associated factors for depression among university students in Ethiopia by their odds ratio, confidence interval, association strength, author, and year of publication.**

| Author, year of publication | Factor category | Associated factors | AOR | 95% CI | Strength of association |
|---|---|---|---|---|---|
| Birhanu et al. 2016 | Socio-demographic factors | Being female | 4.02 | 1.22,4.03 | Strong |
| | | Being first-year student | 3.00 | 3.02,7.25 | Strong |
| | Substance-related | Current use of Khat | 3.05 | 2.05,6.02 | Strong |
| Tamene et al. 2021 | Socio-demographic factors | Female | 9.28 | 3.47–24.81 | Strong |
| | | Academic year | 0.236 | 0.059–0.936 | Weak |
| | | Smoking | 26.3 | 9.33–74.1 | Strong |
| | | Alcohol use | 2.62 | 0.95–7.21 | Strong |
| Ahmed et al. 2020 | Socio-demographic factors | Having monthly pocket money between 500–999 ETB | 0.450 | 0.204–0.995 | Weak |
| | | Being from the College of Social Science and Humanity | 2.582 | 1.332–5.008 | Strong |
| | | Promoted academic performance | 2.912 | 1.063–7.975 | Strong |
| | Clinical factors | Having a mentally ill family member | 2.307 | 1.055–5.049 | Strong |
| | Risky sexual behavior | Having sex after drinking | 3.722 | 1.818–7.619 | Strong |
| | | Being hit by sexual partner | 3.132 | 1.561–6.283 | Strong |
| | Negative life event | Having childhood emotional abuse | 2.167 | 1.169–4.017 | Strong |
| Terasaki et al. 2009 | Negative life event | Moderate outward anger | 1.97 | 1.33–2.93 | Weak |
| | | High outward anger | 3.23 | 2.14–4.88 | Strong |
| | | Violent behavior | 1.82 | 1.37–2.40 | Weak |
| Kebede et al. 2019 | Socio-demographic factors | Age interval 18–21 years | 2.42 | 1.64, 9.22 | Strong |
| | | 1st-year educational level | 1.63 | 1.43, 6.26 | Weak |
| | | 2nd-year educational level, | 1.39 | 1.17, 5.18 | Weak |
| | Negative life event | Stressful life events | 1.61 | 1.14, 2.76 | Weak |
| Muhammed et al. 2019 | Socio-demographic factors | Being male | 1.69 | 1.96–2.98 | Weak |
| | | Study year (first year) | 4.33 | 1.40–13.39 | Strong |
| | Clinical factors | Having a chronic medical illness | 2.07 | 1.19, 3.57 | Strong |
| | | Family history of mental illness | 2.89 | 1.37–6.16 | Strong |
| | Substance use | Khat chewing, | 2.53 | 1.16–5.51 | Strong |
| Dagnew et al. 2020 | Socio-demographic factors | Students who came from a rural family | 1.67 | 1.02–2.72 | Weak |
| | | Those studying Health sciences | 2.65 | 1.34–5.26 | Strong |
| | Clinical factors | Experienced tooth grinding, | 2.79 | 1.36–5.74 | Strong |
| | | Night sleep disturbances | 1.95 | 1.17–3.25 | Weak |
| | | Who reported daytime sleepiness | 1.93 | 1.16–3.20 | Weak |
| | | Stress | 4.20 | 1.90–9.26 | Strong |
| Worku et al. 2020 | Psychosocial | Thinking about a future career prospect | 8.415 | 1.039, 68.14 | Strong |
| Berhanu et al. 2020 | Socio-demographic factors | Female students | 3.36 | 1.88, 6.01 | Strong |
| Teshome et al. 2020 | Socio-demographic factors | Being divorced/widowed | 5.91 | 1.31, 26.72 | Strong |
| | | Being a first-year student | 6.99 | 2.31, 21.15 | Strong |
| | | Being second-year student | 6.25 | 2.05, 19.07 | Strong |
| | | Being a third-year student | 3.85 | 1.26, 11.78 | Strong |
| | Substance-related | Current drinking alcohol | 2.53 | 1.72,3.72 | Strong |
| | | Current smoking cigarettes | 1.71 | 1.02, 2.86 | Weak |
| | | Current use of illicit substances | 2.20 | 1.26, 3.85 | Strong |

of khat [25, 30], alcohol use [26, 27], and having a family history of mental illness [30, 31] were among the most commonly reported factors. The pooled odds ratio of being female among the above-mentioned studies was 5.56, which implies, that those female students were 5.56 times at higher risk of developing depression than males. A meta-analysis study in China showed a

**Table 4. A pooled estimate of the associated factors for depression among students in Ethiopian universities.**

| Associated Factors | Risk Groups | Pooled Effect Size & 95% CI | I2 | Studies Pooled |
|---|---|---|---|---|
| Being female | Students who were females | 5.56 (1.51,9.61) | 99.5% | [25, 26, 28] |
| Being a first-year student | Students who were a first-year student | 4.78 (2.21,7.36) | 99.1% | [25, 27, 30] |
| Current use of khat | Those students who are using Khat | 2.83(2.32,3.33) | 24.6% | [25, 30] |
| Alcohol use | Those students who are using alcohol | 3.12(2.23,4.01) | 83.7% | [26, 27] |
| Family history of mental illness | Students who had a family history of mental illness | 2.57(2.00,3.15) | 46.4% | [30, 31] |

similar conclusion supporting this [49]. Female students are more likely to be depressed [26], and women are more likely than men to suffer from depression. Women are more likely than males to suffer from moderate to severe depression [17]. The disparity could be related to social and cultural factors. Biological conditions are another factor that contributes to the disparity [47].

Besides, the pooled odds ratio of first-year students for the three studies reported above was found to be 4.78. This showed that those who were first-year students were 4.78 times more likely to be depressed than senior students. This might be caused by a lack of social interaction, an unfamiliar exam schedule, a lesser grade than expected, a lack of vacation or a break, a language barrier, or any combination of these factors [50].

Furthermore, the pooled odds ratio of chewing khat and alcohol usage was 2.83 and 3.12 respectively. Even if the cause and effect are not obvious in this study, this result could be related to either the fact that depressed students are more prone to substance use to relieve themselves from the melancholy mood or because maladaptive drug use can modify their mood to the point of depression [1]. Participants who use drugs or alcohol may experience feelings of isolation, despair, and hopelessness that are frequently linked to depression [51].

Finally, students who had a family history of mental illness were 2.57 times more likely to have depression as compared to students who had no family history of mental illness. This association might present as a result of genetic factors, the burden of stigma, and there are many various sorts of financial constraints on family members, and caring for the patient and the children may also put them under stress and worry about their parent's health, which may raise the likelihood that they may experience depression [1]. There are many various sorts of financial constraints on family members, and caring for the patient and the children may also put them under stress and worry about their parent's health, which may raise the likelihood that they may experience depression [52].

## Strengths and limitations

To our knowledge, this is the first meta-analysis of the prevalence of depression among students at Ethiopian universities. However, one of the limitations of this meta-analysis study is that the choice of cut-point by researchers and assessment tools varies depending on where the study was conducted. Second, because so many studies were observational and their subjects were not chosen randomly, it was challenging for us to assess how well they were conducted because so many of them lacked trustworthy information on key factors or appropriate information on the persons they were examining. Confounding and selection bias, therefore, appears inevitable. Thirdly, limited research on mental health in Ethiopia given it is a country that has few psychiatrists nowadays and that stigma may play a role in the responses given by students. Finally, studies other than cross-sectional.

### Implications of this study for clinical practice, researchers, and policymakers

First, this review showed that clinical professionals (clinical psychologists, psychiatrists, sociologists, lecturers, and student counselors) who work in student clinics should be aware that depression is a widespread issue among university students and be prepared to provide patients with management or treatment. Second, the review's findings that the average estimated prevalence of depression among university students is higher than the average estimated prevalence of depression in the general population prompt the question of why this is so and what causes it to be so. Finally, the findings let policy-makers and program planners know that depression is a serious public health issue among university students. This lessens the need for a comprehensive strategy for treating depression among university students.

## Conclusion

This review and meta-analysis study found that the pooled prevalence of depression among students is 28.13%. The findings suggest a high prevalence of depression among university students. Factors being female, being a first-year, chewing khat, alcohol use and family history of mental illness were factors significantly associated with depression.

## Supporting information

**S1 Checklist. PRISMA-P 2015 checklist.**
(DOCX)

**S1 Data.**
(XLSX)

## Acknowledgments

We acknowledge the authors of the included studies for their original contribution.

## Author Contributions

**Conceptualization:** Tamrat Anbesaw, Yosef Zenebe, Moges Gebresellassie.

**Data curation:** Tamrat Anbesaw, Tesfaye Segon, Fasikaw Kebede.

**Formal analysis:** Tamrat Anbesaw, Mogessie Necho, Moges Gebresellassie.

**Methodology:** Tamrat Anbesaw, Yosef Zenebe, Moges Gebresellassie, Tesfaye Segon.

**Software:** Tamrat Anbesaw, Fasikaw Kebede.

**Visualization:** Tamrat Anbesaw, Yosef Zenebe.

**Writing – original draft:** Tamrat Anbesaw, Yosef Zenebe, Mogessie Necho, Fasikaw Kebede.

**Writing – review & editing:** Tamrat Anbesaw, Tesfaye Segon.

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
