## [Decision Letter · Decision Letter 0]

21 Feb 2023

PONE-D-22-34689Prevalence and associated factors of depressive symptoms among Ethiopian University students: a systematic review and meta-analysis.PLOS ONE

Dear Dr. Bete,

Thank you for submitting your manuscript to PLOS ONE. After careful consideration, we feel that it has merit but does not fully meet PLOS ONE’s publication criteria as it currently stands. Therefore, we invite you to submit a revised version of the manuscript that addresses the points raised during the review process.

We look forward to receiving your revised manuscript.

Kind regards,

Wudneh Simegn Belay, MSc

Academic Editor

PLOS ONE

Journal Requirements:

  "The funders had and will not have a role in study design, data collection and analysis, decision to publish, or preparation of the manuscript." 

4. Please amend the manuscript submission data (via Edit Submission) to include author Tamrat Anbesaw.

5. Please amend your authorship list in your manuscript file to include author Tilahun Bete.

Reviewers' comments:

Reviewer's Responses to Questions

**Comments to the Author**

1. Does the manuscript provide a valid rationale for the proposed study, with clearly identified and justified research questions?

Reviewer #1: Partly

Reviewer #2: Partly

Reviewer #3: Yes

2. Is the protocol technically sound and planned in a manner that will lead to a meaningful outcome and allow testing the stated hypotheses?

Reviewer #1: Partly

Reviewer #2: Partly

Reviewer #3: No

3. Is the methodology feasible and described in sufficient detail to allow the work to be replicable?

Reviewer #1: Yes

Reviewer #2: Yes

Reviewer #3: No

4. Have the authors described where all data underlying the findings will be made available when the study is complete?

Reviewer #1: Yes

Reviewer #2: Yes

Reviewer #3: Yes

5. Is the manuscript presented in an intelligible fashion and written in standard English?

Reviewer #1: No

Reviewer #2: Yes

Reviewer #3: No

6. Review Comments to the Author

You may also provide optional suggestions and comments to authors that they might find helpful in planning their study.

Reviewer #1: Title:

" Prevalence and associated factors of depressive symptoms among Ethiopian University students: a systematic review and meta-analysis"

1. Thank you for your invitation to review this systematic review and meta-analysis with topic Prevalence and associated factors of depressive symptoms in Ethiopia: A systematic review and meta-analysis., but my concern there is gross errors on the documents starting from the article type says protocol but is already done article.

2. On abstract section line “26” it says depression which affects 350 million people.no need of magnitude of number here rather talk about the problem and severity. It is better to focus on university students rather than general population.

3. Method line 32, Only three data base is not enough for systematic review and meta-analysis. At list five database is recommended to get adequate number of articles. how do you get access of EMBASE in Ethiopia. Even Addis Ababa university has no access to it, I need strong evidence how they get access? Unless they lay no one had an access in Ethiopia for that data base.

4. What is their search strategy that they follow? it is a mandatory to have search strategy on systematic review and meta-analysis, I could not get nothing on their search strategy.it is not based on PICOT(PICO)… it needs more clarification?

5. Method part of abstract lacks pertinent information about the analysis, Prisma guideline and how they declared the significancy.

6. How many articles on each data base that they got from PubMed, PubMed, Scopus?

7. Conclusion part line,48 says More efforts need to be done to provide better mental healthcare to university students in Ethiopia. It is not your objective conclusion should be based on your finding talk on associated factors that have effect for development of depressive symptoms.

8. Search strategy for each data base is different from one another .so is it the search strategy listed for three of databases? the listed search strategy from line 108 to 117 is not used from PubMed. PubMed is searched with the topic of the review “Prevalence and associated factors of depressive symptoms among Ethiopian University students” how about others like Scopus and Embase?

9. Did this article have registered in any protocol, then what about protocol registration to avoid duplication of efforts?

10. Line 120 inclusion and exclusion criteria it says English language full-text papers were included. What about abstracts that have adequate information on prevalence and predictors.

11. What does this mean “research on non-human subjects”? you are working on university students, or do you have previous knowledge that talks about depression on non-human animals.

12. How do you remove duplicates? on line 126,

13. The PRISMA (preferred reporting items for systematic reviews and meta-analyses) standard was used to perform the literature search. It is a procedure for screening and excluding articles /flow diagram that shows presentation. How it could be used for literature search?

14. Data extraction and appraisal of study quality should be written again it is not clear who should do the interrater disagreement between (TA, and YZ)?

15. Why they used random effect model for Data synthesis and analysis? why not fixed effect?

16. I do not think any article available with The I2 statistical values of zero please see it again.

17. Table 1 heading is too long and difficult to understand better to make it shot and readable.

18. Overall prevalence of depressive symptoms (%) should be recommend to put with confidence interval.

19. On figure -3 after subgroup analysis their high heterogeneity. what do you recommended or expected to be done. I did not see and meta regression analysis.

20. On line 241 Associated factors for depressive symptoms in Ethiopian university … what?

21. No forest plot for the pooled effect estimate/odd ration of factors that have an association on depressive symptoms among university students.

22.” MA” what is this mean?

Reviewer #2: First of all, I would like to thank the journal’s editor(s) for believing in me to review this work. I would also like to thank the authors for conducting such a review. However, the work has several minor flaws, which are provided below.

Title and abstract

First and foremost, is your work a “protocol” or a “full review”?

As can be easily inferred from the title, the authors tried to review the prevalence of the symptoms of depression. However, they did not show which symptom of depression is the most prevalent, and so on. Unless the title should be rephrased as “Prevalence of depression among students at Ethiopian universities and associated factors: a systematic review and meta-analysis.”

Avoid saying “Ethiopian university students,” which implies that only students with Ethiopian citizenship. It might be more acceptable to say "students at Ethiopian universities." Because, grammatically, it makes sense, but its meaning deviates from the aim. Or if the authors' intention was to address only Ethiopian students, they would be right.

“The pooled prevalence of depressive symptoms was 28.13% (95% CI: 22.67, 33.59).” But where are the symptoms? In fact, symptoms, which are only expressed by the person with the condition of interest, are indicators of a disorder. However, the symptoms should be listed in order of their prevalence, as per the aim of the review. We know that depression has various symptoms. This pooled data only showed the prevalence of depression, whether major or minor. The conclusion is ambiguous.

“…one-fourth of students in Ethipisn University…” But the title is different; it is about “Ethiopian university students," not “students in Ethiopian universities.” It is also good to revise grammar issues throughout the manuscript. For instance, because the study is a review covering various universities, it is better to write "universities" than "university." See also the word "Ethipisn" in this phrase.

Background

Page 3 and 4, lines 70-90: This should be discussed in comparing with your finding under the discussion section. In the introduction section, it is better to explain and discuss the extent to which depression impacts the life and activities of students in universities.

Page 3, line 73: “…42.66 of students…” What did you mean?

Page 3, line 82: “The most consistently significantly associated factors among university students…” What did you mean? Please revise grammar issues throughout the manuscript.

Materials and methods

Page 5, lines 110-112: “Our search was conducted on the 10th of the October 2022 using electronic libraries in Scopus, PubMed, and EMBASE and manual exploration of the reference list of articles were the backbones of the current meta-analysis.” This is an ambiguous long sentence and shall be rephrased as “Our search was conducted on October 10, 2022, using electronic libraries in Scopus, PubMed, and EMBASE, as well as manual exploration of the reference lists of articles.”

Page 5, lines 122-124: “The study was cross-sectional, English-language full-text papers, the subject of study should be any type of university student, and studies should be done in Ethiopia.” This sentence needs grammatical revision.

Page, lines 124-127: “Studies published as review articles, qualitative studies, brief reports, letters to the editor, or editorial comments, working papers publications, published in a language other than English, research on non-human subjects, and studies with duplicate data from other studies were also excluded.” Why have you excluded qualitative inquiries while they are important in providing in-depth exploration of the individual students' experiences that could make your systematic review strong evidence to assure the prevalence of depression among your population of interest? Furthermore, why did you use research on non-human subjects as an exclusion criterion when your population of interest was students, i.e., human beings? This is totally out of the scope of your review, so it does not sound like it used animals as an exclusion criterion. The exclusion should be decided within the context of the research goal rather than on the basis of completely separate issues. By the way, could you list those listed as excluded article types?

Page 5, lines 127-128: “The PRISMA (preferred reporting items for systematic reviews and meta-analyses) standard was used to perform the literature search (36).” This is not clear. How did you use the PRISMA checklist for the literature search? As far as I know, they are used to frame the whole review process. The other major issue here is that the PRISMA version you cited is the 2009 version, but you used the PRISMA 2015 version in the supporting file. Why has the discrepancy taken place?

Page 5, lines 130-131: “Two authors (TA, and YZ) checked study titles and abstracts for eligibility after deleting duplicates.” How were the duplicates removed? For example, what automated tool(s) were used to remove the duplicates?

Page 5, lines 131-132: “The full texts were evaluated by the same reviewers if at least one of them thought an article was potentially eligible.” This is also not clear. Why did two authors participate or take part if one of them could decide without consensus?

Page 6, lines 148-149: What is the need to measure the Cochran Q-statistics if I2-indices are employed? Additionally, do you think that these statistics can provide information on the specific factor causing heterogeneity? You said that if I2 is zero, it shows the absence of heterogeneity. Did you think that you were correct or right? If you think so, this is absolutely unacceptable in statistics. Even the included studies have heterogeneity within themselves.

Page 6, lines 150-151: “Subgroup analysis and sensitivity analysis analyses were also used to investigate the source of heterogeneity among the studies included.” This sentence should be rewritten as "Subgroup and sensitivity analyses were also used to investigate the source of heterogeneity among the studies included."

Page 6, line 152: When do you think publication bias occurs?

Results

Page 7, line 166: “…5 articles were obtained from the reference list of included articles…” Could you please cite those articles? It is also recommended to spell out numbers with one digit rather than write them with their digit values.

Page 7, line 169: “…repetitive publications…” How many of them were published repeatedly? Please cite those resources.

Page 7, line 171: “…10 articles were tailored in the final meta-analysis…” How many articles were first included in the qualitative synthesis (the systematic review) before considering eligibility for the quantitative meta-analysis?

Page 7, lines 172-173: “…7 articles were poor methodological assessment, 4 articles were reviewed studies, and 2 articles were published other than the English language).” Please cite them.

Page 8, Table 1: "Tamene et al., 2021" is listed under the heading of the first column in an inappropriate place.

"Seid et al. 2019" is listed as reference number 33. However, even though the source is correct, the reference number 33 is listed differently as "Muhammed et al."

"Berhanu et al. 2020," which is listed in tables 1 and 2 and figures 2–5, is differently cited in reference list number 31.

"Teshome et al. 2019," which is listed as reference 30, is not consistent with its citation; for instance, the year is 2020 in the reference list.

Page 9, lines 202-204: “The pooled prevalence of depressive symptoms among university students from ten studies (28-35, 43, 44) included studies conducted in Ethiopia was 28.13% (95% CI: 22.67, 33.59) with significant heterogeneity among the studies (I2 = 100%, p-value < 0.001) (Fig 2).” It needs grammar revision.

Page 10, lines 230-231: “However, the result showed that there was no single influential study since the 95% CI interval result was obtained when each of the ten studies was excluded at a time (Fig 5).” It is not clear.

Page 12, lines 161-162: Please avoid the use of words such as "furthermore" and "also” simultaneously.

Discussion

Please do not repeat strategies, and procedures under this section, which you mentioned in the method’s section. Simply discuss your findings by comparing them to other relevant resources. For instance, the explanation from lines 274–282 is unnecessary.

Page 13, lines 268-272: This shall be discussed under the discussion's sub-title, "practical implications."

Page 14, lines 288-292: Is it appropriate to use "additionally" with "also" or "furthermore" with "also" simultaneously?

Page 14, line 294: “The findings revealed that several distinct characteristics…” Which findings? Yours or the comparators you mentioned? It is not clear.

Page 15, line 319: “…were were…” Please delete one.

Page 16, lines 344-347: Where did you get this? It needs to be cited. Or you should not be sure if it is your own justification.

Strengths and limitations

Page 16, line 349: Write "MA" in its expanded form, or you can abbreviate it from the beginning.

Can including old references be a strength of your review while our world is being drastically changed by technology, which is also a major reason for students to be frustrated with their education, which could result in anxiety and depression during exam time, and so on?

Please include other limitations such as pooling data despite high heterogeneity, studies other than cross-sectional, studies published in other languages, etc. Because all of these are the limitations of this review.

Implications of this study for clinical practice, researchers, and policymakers

Why did you skip the role of psychologists, psychiatrists, sociologists, lecturers, student counselors, etc. working in universities, as they are not stakeholders to take part in solving the issue?

Conclusion

Please be focused on the prevalence and the associated factors. Avoid making recommendations in this section, as they have already been mentioned under the implications section.

Declarations

Is the review protocol not registered in a registry data base, such as PROSPERO? If not, how did you know your review is the first of its kind in Ethiopia?

References

There are several major flaws within the organization of the reference lists. So, you should revise cautiously.

Reviewer #3: The authors should work more on methodology, errors and coherence of sentences.

It seems as you followed PRISMA checklist for reporting this systematic review and meta-analysis but you did not followed it correctly. Please follow The PRISIMA checklist while reporting and the whole method should be reorganized based on PRISMA checklist recommendation.

7. PLOS authors have the option to publish the peer review history of their article (what does this mean?). If published, this will include your full peer review and any attached files.

Reviewer #1: No

Reviewer #2: **Yes: **Ewunetie Mekashaw Bayked

Reviewer #3: No

---

## [Author Response · Author response to Decision Letter 0]

10 May 2023

Comments to the authors

PLOS ONE

PONE-D-22-34689

Article Type: Study Protocol

Prevalence and associated factors of depressive symptoms among Ethiopian University students: a systematic review and meta-analysis

First of all, I would like to thank the journal’s editor(s) for believing in me to review this work. I would also like to thank the authors for conducting such a review. However, the work has several minor flaws, which are provided below.

#Response: Thank you for your positive feedback, it gives the strength do more.

Title and abstract

First and foremost, is your work a “protocol” or a “full review”?

#Response: Thank you, dear reviewer. We will update the system which a full review.

As can be easily inferred from the title, the authors tried to review the prevalence of the symptoms of depression. However, they did not show which symptom of depression is the most prevalent, and so on. Unless the title should be rephrased as “Prevalence of depression among students at Ethiopian universities and associated factors: a systematic review and meta-analysis.”

#Response: Thank you dear reviewer for your positive feedback. We accepted and modified it.

Avoid saying “Ethiopian university students,” which implies that only students with Ethiopian citizenship. It might be more acceptable to say "students at Ethiopian universities." Because, grammatically, it makes sense, but its meaning deviates from the aim. Or if the authors' intention was to address only Ethiopian students, they would be right. 

#Response: Again we appreciate your comment. Amended.

“The pooled prevalence of depressive symptoms was 28.13% (95% CI: 22.67, 33.59).” But where are the symptoms? In fact, symptoms, which are only expressed by the person with the condition of interest, are indicators of a disorder. However, the symptoms should be listed in order of their prevalence, as per the aim of the review. We know that depression has various symptoms. This pooled data only showed the prevalence of depression, whether major or minor. The conclusion is ambiguous. 

#Response: Thank you for your comment. We modified the title based on your recommendation. We appreciate your comments. See the revised manuscript.

“…one-fourth of students in Ethipisn University…” But the title is different; it is about “Ethiopian university students," not “students in Ethiopian universities.” It is also good to revise grammar issues throughout the manuscript. For instance, because the study is a review covering various universities, it is better to write "universities" than "university." See also the word "Ethipisn" in this phrase.

#Response: We completely accepted and action has been taken. Thank you so much.

Background

Page 3 and 4, lines 70-90: This should be discussed in comparing with your finding under the discussion section. In the introduction section, it is better to explain and discuss the extent to which depression impacts the life and activities of students in universities. 

#Response: Thank you we rewrite again. We minimized it.

Page 3, line 73: “…42.66 of students…” What did you mean? 

#Response: Typing error. We want to describe the percentage of depression which is 42.66 %.

Page 3, line 82: “The most consistently significantly associated factors among university students…” What did you mean? Please revise grammar issues throughout the manuscript. 

#Response: We amended it as “Different factors significantly associated with depression among university students such as….”

Materials and methods

Page 5, lines 110-112: “Our search was conducted on the 10th of the October 2022 using electronic libraries in Scopus, PubMed, and EMBASE and manual exploration of the reference list of articles were the backbones of the current meta-analysis.” This is an ambiguous long sentence and shall be rephrased as “Our search was conducted on October 10, 2022, using electronic libraries in Scopus, PubMed, and EMBASE, as well as manual exploration of the reference lists of articles.”

#Response: Really dear reviewer we appreciate you help to amend our manuscript. We revised as recommended.

Page 5, lines 122-124: “The study was cross-sectional, English-language full-text papers, the subject of study should be any type of university student, and studies should be done in Ethiopia.” This sentence needs grammatical revision. 

#Response: We revised it.

Page, lines 124-127: “Studies published as review articles, qualitative studies, brief reports, letters to the editor, or editorial comments, working papers publications, published in a language other than English, research on non-human subjects, and studies with duplicate data from other studies were also excluded.” Why have you excluded qualitative inquiries while they are important in providing in-depth exploration of the individual students' experiences that could make your systematic review strong evidence to assure the prevalence of depression among your population of interest? Furthermore, why did you use research on non-human subjects as an exclusion criterion when your population of interest was students, i.e., human beings? This is totally out of the scope of your review, so it does not sound like it used animals as an exclusion criterion. The exclusion should be decided within the context of the research goal rather than on the basis of completely separate issues. By the way, could you list those listed as excluded article types?

#Response: Thank you for your constructive comment. This was our predetermined exclusion criteria. We have not excluded qualitative studies during the actual analysis. We that some articles conducted on human subject will be obtained during the search process. But after searching process we have nor get such articles. We removed “research on non-human subjects”. 

Page 5, lines 127-128: “The PRISMA (preferred reporting items for systematic reviews and meta-analyses) standard was used to perform the literature search (36).” This is not clear. How did you use the PRISMA checklist for the literature search? As far as I know, they are used to frame the whole review process. The other major issue here is that the PRISMA version you cited is the 2009 version, but you used the PRISMA 2015 version in the supporting file. Why has the discrepancy taken place?

#Response: We thank your critical point of view. The PRISMA (preferred reporting items for systematic reviews and meta-analyses) standard was used to perform the systematic reviews and meta-analyses process. But during the literature search process. We cite the updated reference (PRISMA ). We accepted your very supportive comment. Really we are surprising by your comment.

Page 5, lines 130-131: “Two authors (TA, and YZ) checked study titles and abstracts for eligibility after deleting duplicates.” How were the duplicates removed? For example, what automated tool(s) were used to remove the duplicates? 

#Response: Using the endnote reference manager duplicated articles were excluded. Again we revised the sentence. See document.

Page 5, lines 131-132: “The full texts were evaluated by the same reviewers if at least one of them thought an article was potentially eligible.” This is also not clear. Why did two authors participate or take part if one of them could decide without consensus?

#Response: We accepted your constructive comments. See the main document.

Page 6, lines 148-149: What is the need to measure the Cochran Q-statistics if I2-indices are employed? Additionally, do you think that these statistics can provide information on the specific factor causing heterogeneity? You said that if I2 is zero, it shows the absence of heterogeneity. Did you think that you were correct or right? If you think so, this is absolutely unacceptable in statistics. Even the included studies have heterogeneity within themselves. 

 

#Response: We accepted your comment. The Higgs I2 statistic was also utilized to detect heterogeneity. 

Page 6, lines 150-151: “Subgroup analysis and sensitivity analysis analyses were also used to investigate the source of heterogeneity among the studies included.” This sentence should be rewritten as "Subgroup and sensitivity analyses were also used to investigate the source of heterogeneity among the studies included."

#Response: We accepted it.

Page 6, line 152: When do you think publication bias occurs?

#Response: Publication bias occurs when the publication of research results depends not just on the quality of the research but also on the hypothesis tested, and the significance and direction of effects detected. During testing meta-bias test, the P- value is < 0.05.

Results

Page 7, line 166: “…5 articles were obtained from the reference list of included articles…” Could you please cite those articles? It is also recommended to spell out numbers with one digit rather than write them with their digit values. 

#Response: We Accepted and cited it.

Page 7, line 169: “…repetitive publications…” How many of them were published repeatedly? Please cite those resources. 

#Response: This is a very critical view and we accept the comments. This editorial problem. repetitive publications is replacedby duplicated articles. There was too many duplication. In the future we will consider your comment.

Page 7, line 171: “…10 articles were tailored in the final meta-analysis…” How many articles were first included in the qualitative synthesis (the systematic review) before considering eligibility for the quantitative meta-analysis? 

#Response: Full-text articles excluded with reasons (n = 13, 7 poor methodological assessment, and 4 were (n = 13, 7 poor methodological assessment, and 4 were reviewed studies, and 2 were published other than English language)

Response: We amended it as per recommendation.

Page 7, lines 172-173: “…7 articles were poor methodological assessment, 4 articles were reviewed studies, and 2 articles were published other than the English language).” Please cite them. 

#Response: Thank you, dear reviewer. We accepted the comment.

Page 8, Table 1: "Tamene et al., 2021" is listed under the heading of the first column in an inappropriate place. 

#Response: Thank you for your critical view. We removed it.

"Seid et al. 2019" is listed as reference number 33. However, even though the source is correct, the reference number 33 is listed differently as "Muhammed et al." 

#Response: Revised.

"Berhanu et al. 2020," which is listed in tables 1 and 2 and figures 2–5, is differently cited in reference list number 31. 

#Response: We have seen it and are now correctly cited.

"Teshome et al. 2019," which is listed as reference 30, is not consistent with its citation; for instance, the year is 2020 in the reference list.

#Response: We accepted and revised accordingly.

Page 9, lines 202-204: “The pooled prevalence of depressive symptoms among university students from ten studies (28-35, 43, 44) included studies conducted in Ethiopia was 28.13% (95% CI: 22.67, 33.59) with significant heterogeneity among the studies (I2 = 100%, p-value < 0.001) (Fig 2).” It needs grammar revision.

#Response: We revised it accordingly. Thank you dear reviewer. Please see the main document.

Page 10, lines 230-231: “However, the result showed that there was no single influential study since the 95% CI interval result was obtained when each of the ten studies was excluded at a time (Fig 5).” It is not clear. 

#Response: We write it as “The findings show that all values are within the estimated 95% confidence interval, indicating that the absence of one study had no significant difference on the prevalence of this meta-analysis.”

Page 12, lines 161-162: Please avoid the use of words such as "furthermore" and "also” simultaneously.

#Response: We removed it.

Discussion 

Please do not repeat strategies, and procedures under this section, which you mentioned in the method’s section. Simply discuss your findings by comparing them to other relevant resources. For instance, the explanation from lines 274–282 is unnecessary.

#Response: Removed. Thank you for your comment!

Page 13, lines 268-272: This shall be discussed under the discussion's sub-title, "practical implications." 

#Response: We accepted.

Page 14, lines 288-292: Is it appropriate to use "additionally" with "also" or "furthermore" with "also" simultaneously?

#Response: Again we appreciated for your critical view. 

Page 14, line 294: “The findings revealed that several distinct characteristics…” Which findings? Yours or the comparators you mentioned? It is not clear.

#Response: Dear reviewer we included it for comparison. Being a first-year, substance use like chewing khat, family history of mental illness and other several factors can contribute for the risk of depression compared to general population. Several characteristics of university students, such as increased social interactions and shifting residential and financial situations, may raise the risk of depression.

Page 15, line 319: “…were were…” Please delete one. 

#Response: Removed

Page 16, lines 344-347: Where did you get this? It needs to be cited. Or you should not be sure if it is your own justification.

#Response: We cited it.

Strengths and limitations 

Page 16, line 349: Write "MA" in its expanded form, or you can abbreviate it from the beginning.

Can including old references be a strength of your review while our world is being drastically changed by technology, which is also a major reason for students to be frustrated with their education, which could result in anxiety and depression during exam time, and so on? 

#Response: Based on your valuable comment. We avoided from the strength of the study. We revised it. Our assumption is not to exclude findings from the study.

Please include other limitations such as pooling data despite high heterogeneity, studies other than cross-sectional, studies published in other languages, etc. Because all of these are the limitations of this review. 

#Response: We included in the review.

Implications of this study for clinical practice, researchers, and policymakers

Why did you skip the role of psychologists, psychiatrists, sociologists, lecturers, student counselors, etc. working in universities, as they are not stakeholders to take part in solving the issue?

#Response: Thank you! the role for treating of disorder is clinical professionals that include clinical psychologists and psychiatrists and we rewrite it comprehensively as per showing the direction. We incorporated it.

Conclusion 

Please be focused on the prevalence and the associated factors. Avoid making recommendations in this section, as they have already been mentioned under the implications section.

Response: We amended as per recommendation.

Declarations

Is the review protocol not registered in a registry data base, such as PROSPERO? If not, how did you know your review is the first of its kind in Ethiopia?

Response: We have conducted an extensive search strategy with all search data bases. Not obtained systematic review and meta-analysis were conducted among students. 

References

There are several major flaws within the organization of the reference lists. So, you should revise cautiously.

Response: We revised the thoroughly. Thank you for your all positive comments. 

Reviewer 2

An important topic was raised in this review and meta-analysis even though meta-analysis in prevalence studies are hard to interpret and generalize because of considerable heterogeneity. I presented here my comments which helped the authors to strengthen their manuscript. 

Response: Thank you for your positive feedback, it gives the strength to do more.

Abstract 

Result: Change the sentence data was extracted…. to data were extracted…

Response: Thank you, dear reviewer. We revised it.

Studies that utilized other (PHQ-9, HADS); 30.67% higher than studies that used BDI-II; 26.07 %. Not clear please reorganize this sentence. 

Response: Thank you dear reviewer. The measurement tools for depressive symptoms to perform subgroup analysis. The subgroup analysis by assessment instrument yields that measurement with other (PHQ-9, HADS), CEDS’s, and BDI-II. We categorized it for subgroup analysis and the result showed depression assessed using (PHQ-9, HADS) higher than studies BDI-II.

Change aOR to AOR

Conclusion: change Ethipisn to Ethiopia

Response: Thank you for your critical review.

Background 

Paragraph 1: guilty feeling repeated twice

Response: We took a measurement.

Paragraph 3: Pakistani to Pakistan

Response: Thank you so much for your critical view.

Paragraph 4: In the context of Ethiopia too….not clear please reconstruct the sentence

Response: We rewrite it again.

Materials and method 

It seems as you followed PRISMA checklist for reporting this systematic review and meta-analysis but you did not followed it correctly. Please follow The PRISIMA checklist while reporting and the whole method should be reorganized based on PRISMA checklist recommendation. 

Response: Based on your valuable comment. We have reviewed the whole manuscript using PRISMA 2015 checklist. 

Please state the searching period correctly from when to when. You write a specific time period i.e. 10th of the October 2022.

Response: Thank you we revised it based on your great view of point. Our search was conducted on October 10, 2022, using electronic libraries in Scopus, PubMed, and EMBASE, as well as manual exploration of the reference lists of articles. We searched for any additional articles until the 10th of October 2022. Under Inclusion Criteria, we added this sentence “All observational studies were conducted by using different study-designed cross-sectional reports from June 2006 up to June 2021 were included.”

In your inclusion and exclusion criteria you stated as studies ………published in a language other than English were excluded. Do you believe that studies conducted other than English language can be conducted and published in Ethiopia? I think there is no study conducted in any other language so please remove this. 

Response: Very amazing point of view. We accepted and removed it.

The PRISMA (preferred reporting items for systematic reviews and meta-analyses) standard was used to perform the literature search. This sentence should be removed from this and written as a study protocol before search strategy. 

Response: Thank you which is another very constructive suggestion. We amended as per recommendation and see the revised manuscript.

You extract the detailed data using a standardized spreadsheet, what are the detailed data’s. It should be listed.

Response: We revised it exhaustively dear reviewer. Please see the revised document.

Since all the studies included for this systematic review and meta-analysis were crossectional studies, you will be used Newcastle Ottawa quality assessment scale adapted for crossectional studies to assess the quality of the studies rather than Newcastle Ottawa quality evaluation checklist. So you should to assess the quality of each study based on Newcastle Ottawa quality assessment scale adapted for crossectional studies again. 

Response: We have assessed the quality again. Thank you.

What is your outcome measurement? The outcome measurement should be explicitly stated before Data synthesis and analysis.

Response: Thank you very much. Based on your valuable comment. We incorporated it.

Outcome measurements

We have two objectives in this systematic review and meta-analysis study. These are to determine the pooled prevalence of depression among university students in Ethiopia and to estimate the pooled effects of associated factors with depression among university students in Ethiopia. The pooled prevalence of depression was calculated using STATA version 14.0. The pooled effect estimate of associated factors with depression was calculated. The odds ratio was prepared from the searched research reports using two by two tables.

Data analysis 

You stated as you used Higgs I2 statistic and Cochran Q-statistics were also utilized to detect heterogeneity. But you did not use Cochran Q-statistics? Why?

Response: Cochran's Q test is a non-parametric statistical test to verify whether k treatments have identical effects. We revised it accordingly. We thank you! 

The I2 statistical values of zero, 25, 50, and 75% connote absence, little, moderate, and great heterogeneity, respectively. This is vague please categorize it. 

Response: We revised it accordingly. “Thus, percentages I2 statistical values around 0%( I2 0), 25% (I2 25), 50% (I2 50), and 75% (I2 75) would mean absent, low, medium, and high heterogeneity, respectively.” We cited it.

Results 

Your systematic review and meta-analysis is conducted in Ethiopia and your search will be studies conducted in Ethiopia. How 2 articles were removed because published other than the English language. 

Response: We took the action. Thank you so much 

There is a considerable heterogeneity I2 =100%, so you tried to explain the source of heterogeneity using subgroup analysis considering the sample size of each study and the study instrument. Even the source of heterogeneity is not explained, I2 =100%. If the source of heterogeneity is not explained, what you would do? Why you will limit subgroup analysis on sample size and the study instrument 

Response: We appreciate all comments, which give an additional body of knowledge. We used a sample size and instrument for subgroup analysis rather than case for depression. 

Regards,

---

## [Decision Letter · Decision Letter 1]

14 Jun 2023

PONE-D-22-34689R1Prevalence of depression among students at Ethiopian Universities and associated factors: a systematic review and meta-analysis.PLOS ONE

Dear Dr. Anbesaw,

Thank you for submitting your manuscript to PLOS ONE. After careful consideration, we feel that it has merit but does not fully meet PLOS ONE’s publication criteria as it currently stands. Therefore, we invite you to submit a revised version of the manuscript that addresses the points raised during the review process.

Avoid Email addresses except the correspondence author in the title page.

Tamrat Anbesaw, Wollo University, College of Medicine and Health 26 Sciences, Department of Psychiatry, Dessie, Ethiopia, should be revised as “*Corresponding author”.

-Authors’ contributions should be prepared based on the journal guideline.

We look forward to receiving your revised manuscript.

Kind regards,

Wudneh Simegn Belay, MSc

Academic Editor

PLOS ONE

Journal Requirements:

Additional Editor Comments:

Reviewer #1: Dear authors thank you for accepting and revising this article based on the comments that i raised including the comments from other reviewers. i hope this is an excellent finding for researchers and policy makers who are working on the field of mental health especially among university students. it is helpfull for intervention and planning prior actions before students who join university.

withregards

Reviewer #3: Thank you. All my comments are addressed appropriately.

---

## [Author Response · Author response to Decision Letter 1]

15 Jun 2023

PLOS ONE

PONE-D-22-34689

Article Type: Study Protocol

Title: Prevalence and associated factors of depressive symptoms among Ethiopian University students: a systematic review and meta-analysis

Thank you dear Editor, and Reviewers for your great contribution. We will address the following issues ;

1. Avoid Email addresses except the correspondence author in the title page.

Tamrat Anbesaw, Wollo University, College of Medicine and Health 26 Sciences, Department of Psychiatry, Dessie, Ethiopia, should be revised as “*Corresponding author”.

 Response: We removed it.

2. Authors’ contributions should be prepared based on the journal guideline.

Response: We formatted the manuscript using the PLoS ONE style. 

Authors contributions

Conceptualization: Tamrat Anbesaw, Yosef Zenebe.

Data curation: Tamrat Anbesaw, Yosef Zenebe, Mogessie Necho,Tesfaye Segon.

Formal analysis: Tamrat Anbesaw, Yosef Zenebe

Funding acquisition: Tamrat Anbesaw, Yosef Zenebe.

Investigation: Tamrat Anbesaw, Yosef Zenebe.

Methodology: Tamrat Anbesaw, Yosef Zenebe, Mogessie Necho, Moges Gebresellassie,Tesfaye Segon, Fasikaw Kebede, Tilahun Bete

Project administration: Tamrat Anbesaw, Yosef Zenebe 

Resources: Tamrat Anbesaw, Yosef Zenebe, Mogessie Necho, Moges Gebresellassie,Tesfaye Segon, Tilahun Bete

Software: Tamrat Anbesaw, Yosef Zenebe 

Validation: Tamrat Anbesaw, Yosef Zenebe

Writing – original draft: Tamrat Anbesaw

Writing – review & editing: Tamrat Anbesaw, Yosef Zenebe, Mogessie Necho, Tesfaye Segon, Fasikaw Kebede, Tilahun Bete

---

## [Editor Report · Decision Letter 2]

2 Jul 2023

Prevalence of depression among students at Ethiopian Universities and associated factors: a systematic review and meta-analysis.

PONE-D-22-34689R2

Dear Dr. Anbesaw,

We’re pleased to inform you that your manuscript has been judged scientifically suitable for publication and will be formally accepted for publication once it meets all outstanding technical requirements.

Kind regards,

Wudneh Simegn Belay, MSc

Academic Editor

PLOS ONE
---

## [Editor Report · Acceptance letter]

21 Aug 2023

PONE-D-22-34689R2 

Prevalence of depression among students at Ethiopian Universities and associated factors: a systematic review and meta-analysis 

Dear Dr. Anbesaw:

I'm pleased to inform you that your manuscript has been deemed suitable for publication in PLOS ONE. Congratulations! Your manuscript is now with our production department. 

Kind regards, 

on behalf of

Dr. Wudneh Simegn 

Academic Editor

PLOS ONE